# Risk Factors for Exercise-Associated Sudden Cardiac Death in Thoroughbred Racehorses

**DOI:** 10.3390/ani12101297

**Published:** 2022-05-18

**Authors:** Laura Nath, Andrew Stent, Adrian Elliott, Andre La Gerche, Samantha Franklin

**Affiliations:** 1School of Animal and Veterinary Sciences, University of Adelaide, Roseworthy 5371, Australia; sam.franklin@adelaide.edu.au; 2Faculty of Veterinary and Agricultural Sciences, University of Melbourne, Werribee 3030, Australia; awstent@gmail.com; 3Centre for Heart Rhythm Disorders, University of Adelaide, Adelaide 5005, Australia; adrian.elliott@adelaide.edu.au; 4Baker Heart and Diabetes Institute, Melbourne 3004, Australia; andre.lagerche@baker.edu.au

**Keywords:** sudden cardiac death, arrhythmia, horse, fatality, risk factors, training, racing, post-mortem, athlete

## Abstract

**Simple Summary:**

Understanding the causes of racehorse deaths is an important step in addressing this welfare concern. Previous studies have identified that during Thoroughbred racing, catastrophic musculoskeletal injuries, which necessitate euthanasia, account for approximately 75% of fatalities, with sudden athletic death occurring in the remaining 25% of cases. In sudden athletic death cases a post-mortem examination is needed to determine the cause of death. Approximately half of these sudden athletic deaths are attributed to a fatal cardiac arrhythmia, which is termed sudden cardiac death. In order to investigate risk factors for sudden cardiac death, we reviewed the post-mortem reports from horses that died on metropolitan racetracks in Melbourne, Australia and grouped horses into sudden cardiac death and all other fatal injuries. We found that horses with sudden cardiac death were more likely to die during training than during racing, had fewer lifetime starts and were less likely to be entire (uncastrated) males. Exercise intensity appears not to be critically important in precipitating sudden cardiac death in horses. Sudden cardiac death occurred early in the careers of affected horses.

**Abstract:**

Cardiac arrhythmias resulting in sudden cardiac death (SCD) are an important cause of racehorse fatalities. The objective of this study was to determine risk factors for SCD in Thoroughbreds by evaluating a sample with a policy of mandatory post-mortem following racing or training fatalities. Risk factors were compared between case horses with SCD (*n* = 57) and control horses with other fatal injury (OFI, *n* = 188) by univariable and multivariable logistic regression. Survival in years for horses with SCD was compared to OFI using the Kaplan–Meier method with log rank test. The following variables were most important in the multiple logistic model: Horses with SCD were more likely to die during training than during racing, SCD (42/57, 74%) vs. OFI (82/188, 44%; odds ratio [OR], 95% confidence interval [CI], 2.5, 1.2–5.4; *p* = 0.01), had fewer lifetime starts, median (interquartile range [IQR]), SCD (3.0 [0.0–9.0]) vs. OFI (9.0 [0.0–22.8]; OR, 95% CI, 0.96, 0.9–1.0; *p* = 0.02 and were less likely to be entire (uncastrated) males, SCD 9/57 (16%) vs. OFI (46/188, 25%; OR, 95% CI, 0.47, 0.1–0.9; *p* = 0.03). Survival in years (median (IQR)) for horses with SCD was 3.6 (3.1–4.4), which was shorter than OFI (4.5 [3.1–6.0], hazard ratio, 95%CI, 1.6,1.2–2.3; *p* < 0.001). SCD occurs more commonly in training than racing, which suggests exercise intensity is less important in precipitating this fatality. In this study, SCD occurred early in the careers of affected horses.

## 1. Introduction

Fatal cardiac arrhythmia resulting in sudden cardiac death (SCD) is an important cause of horse fatalities during racing and training [1,2]. Dying suddenly in association with exercise is termed sudden athletic death (SAD) [3]. In Thoroughbred racehorses, suspected cardiac disease accounts for approximately 55% of sudden athletic deaths and approximately 12% of all race day fatalities [4,5,6,7]. Even with thorough post-mortem examination, SCD is challenging to diagnose. Since cardiac arrhythmia is an electrical event, cases of SCD often have negative or equivocal findings on post-mortem studies [5,8,9]. Pulmonary haemorrhage, congestion and oedema are commonly observed during post-mortem examination of horses that have died during or immediately following exercise, but these findings are of uncertain significance as a cause of death and may occur secondary to cardiac failure [5,6,8,10].

In humans, fatal cardiac arrhythmia associated with underlying cardiovascular disease is the most common mechanism of cardiac arrest and death during sports [11]. Inherited or congenital cardiac diseases, including inherited cardiomyopathies and ion channelopathies, present almost exclusively in younger athletes and encompass the majority of SCDs in this group [11]. Therefore, these are the primary cause of SCD in young athletes in the absence of acquired cardiovascular disease or training induced cardiovascular abnormalities. In human athletes, structural and electrical cardiac remodelling accompanies increased volumes of exercise training but their significance in the pathogenesis of arrhythmias and SCD is not clear [12,13]. The underlying pathology associated with SCD is not well established in horses, although myocardial inflammation and fibrosis, chamber dilation and trabecular and myocardial hypertrophy have been described [1,5,9].

Increased age and race distance, along with racing in summer, have been identified as risk factors for race-associated SAD in Thoroughbred horses [14]. These risk factors are shared with catastrophic musculoskeletal injury [15,16,17]. In Norwegian and Swedish harness horses, an increased incidence of SAD was seen in spring [10]. This study also found an increased risk of SAD in horses that raced more frequently in the 30 and 180 days prior to the fatality [10]. In contrast, in Thoroughbred racehorses, an increased risk of SAD was found in horses that raced less frequently in the 60 days prior to the fatality [14]. Cumulative volume of training and racing are likely to increase the risk of fatality associated both with SCD and musculoskeletal injury. In human athletes, structural and electrical cardiac remodelling accompanies increased volumes of exercise training but their significance in the pathogenesis of arrhythmias and SCD is not clear [12,13]. The underlying pathology associated with SCD is not well established in horses, although myocardial inflammation and fibrosis, chamber dilation and trabecular and myocardial hypertrophy have been described [1,5,9].

Approximately 53–60% of sudden death fatalities in racehorses and horses engaged in other sport disciplines occur in training, supporting the concept that fatalities regularly occur at sub-maximal exercise intensity [9,18]. Risk factor studies incorporating data from training fatalities are limited to investigation of catastrophic musculoskeletal injury [16,19]. Risk factors for SCD, rather than sudden death [14], or sudden athletic death [10], as a subset of exercise-associated fatalities have not been previously explored.

The hypothesis of this study was that risk factors for SCD will be distinct from other fatal injuries when cases from both racing and training are included. The objectives of this study were to explore associations between SCD and potentially relevant risk factor variables that were able to be extracted from the public record or veterinary records. A secondary objective was to investigate the age distribution of horses with SCD compared to the registered racing population in Australia.

## 2. Materials and Methods

### 2.1. Study Design and Data Collection

A retrospective case-control study of Thoroughbreds racing and training in Melbourne, Australia, between 27 April 2007 and 9 March 2021 was conducted. Post-mortem reports pertaining to all fatalities during the period were obtained with permission from the veterinary department of the governing body (Racing Victoria). For named horses, date of birth, country of origin of each horse and the sire and dam of each horse were obtained from the Racing Australia [20] database. For unnamed horses, the horse was identified by the brands, microchip and reported parentage in the post-mortem report and confirmed by review of the Australian Studbook database [21]. The studbook database was then used to obtain the date of birth and country of origin of horse, sire and dam. The sex of the horse, date of fatality and exercise type at the time of death were obtained from the post-mortem report and confirmed by reviewing the racing record on the Racing Australia database [20]. Lifetime starts were obtained from the Racing Australia database [20].

### 2.2. Case Selection, Case Definition and Inclusion Criteria

All fatalities of Thoroughbred racehorses occurring on metropolitan racetracks or training facilities in Melbourne, Australia, are subject to mandatory post-mortem at a single facility (University of Melbourne, Werribee, Australia). Fatalities occurring on other regional tracks are submitted for post-mortem on a discretionary basis at the request of attending veterinarians or stewards. To limit potential bias, only metropolitan cases were included in the study.

Post-mortem reports were reviewed by a single author (LN) and cases were grouped into SCD or other fatal injury (OFI) or excluded from the study. Cases were excluded according to the following criteria; non-exercise associated death, non-metropolitan fatality, non-racehorse, identity of horse not confirmed. Cases in which the horse was subjected to euthanasia due to post-exercise distress without catastrophic injury were also excluded. Exercise-associated deaths were defined as sudden death occurring within one hour of racing or training or euthanasia within 48 h due to a catastrophic injury sustained during racing or training. The SCD group comprised all horses in which the definitive cause of death was attributed to cardiac pathology or in which fatal cardiac arrhythmia could not reasonably be excluded as the cause of death. Cases with a diagnosis of pulmonary oedema, congestion and haemorrhage were included in the SCD group. The control group consisted of all other cases of fatality that were subject to euthanasia due to a catastrophic injury or had sudden death that could not be attributed to cardiac pathology.

### 2.3. Data Analysis

#### 2.3.1. Software

Statistical analyses were performed in Graphpad Prism version 9^a^ (San Diego, CA, USA). Statistical significance was set as *p* < 0.05. Confidence intervals for proportions were calculated using Graphpad QuickCalcs^b^ (San Diego, CA, USA).

#### 2.3.2. Risk Factor Analysis

Potential risk factors were evaluated using univariable and multivariable logistic regression. Variables were selected based on whether they were biologically plausible or previously identified in the literature. The possible risk factors were at horse-level; age, sex, hemisphere of origin of horse, sire and dam and environmental-level; exercise type prior to death, lifetime starts and season. Age at the time of death was determined for each horse by calculating the number of days between birth and death, measured as a continuous variable. Sex was categorised as female, gelding or entire (uncastrated) male and examined as separate categorical variables. As a proxy for pedigree, the origin of each horse, sire and dam was determined and categorised by hemisphere of origin into northern and southern hemisphere. Exercise type preceding death was categorised as either training or racing. Metropolitan flat or jump racing starts or official trials were included as racing fatalities. Potential risk factors were screened using univariable logistic regression and odds ratio (OR) reported alongside 95% confidence intervals (95% CI). Variables with an F test value of *p* ≤ 0.25 were included in the multivariable logistic regression model. Lifetime starts were recorded as 0 for unraced horses and measured as a continuous variable. Continuous variables were tested for normality with the Shapiro–Wilk test. Non-normally distributed variables were reported as median and interquartile range (IQR). Confidence intervals for proportions were calculated by the modified Wald method. The correlation between age and lifetime starts was investigated with the Spearman rank coefficient.

#### 2.3.3. Survival Analysis

Survival in years was assessed using the Kaplan–Meier method. Comparison of survival for horses with SCD to those with OFI was performed by applying a log-rank (Mantel–Cox) test and the hazard ratio was derived by the log rank method. Adjustment for non-proportional hazards was made using the life expectancy difference (LED) and life expectancy ratio (LER) [22].

#### 2.3.4. Comparison to Horses in the Australian Racing Population

The age of individual horses racing for each year from 2007–2020 was obtained from the Racing Australia Factbook [23]. The racing age in years, based on a birthday of 1st August, rather than the calendar age was used. The frequency distribution of age (≤2, 3, 4, 5, 6, 7, ≥8) was expressed as a percentage of all racing horses, and compared to the frequency distributions of SCD case horses and OFI case horses using a one-way ANOVA and Kruskal–Wallis test for multiple comparisons.

## 3. Results

### 3.1. Animals

There were 57 case horses with SCD and 188 control horses with OFI retained within the study, details of which are provided in Figure 1. All were Thoroughbred racehorses, median (IQR) age 4.2 (3.1–5.6) years, with 69 females, 121 geldings and 55 entire males.

### 3.2. Post-Mortem Classification

A flow chart summarising classification of cases is presented in Figure 1. Post-mortems were performed at a single institution by one of 19 qualified veterinary pathologists. The horses were classified according to the cause of death attributed by the attending pathologist. The diagnoses provided by the attending pathologists in the 57 horses with a final classification of SCD were as follows: 26 horses had a diagnosis of SCD without any other significant lesions, three horses had SCD and exercise-induced pulmonary haemorrhage (EIPH), two horses had SCD and pulmonary oedema and congestion, two horses had SCD and pulmonary oedema and one horse had SCD with EIPH, pulmonary oedema and congestion. Of the six horses with myocardial lesions, three horses had a final diagnosis of cardiomyopathy, two horses had myocarditis and one horse had left ventricular concentric hypertrophy. Of the remaining 17 horses in which SCD could not reasonably be excluded, 11 horses had severe EIPH, two horses had mild EIPH, three horses had EIPH and pulmonary oedema and congestion, and one horse had pulmonary oedema alone.

The 188 horses classified with OFI included 183 horses that were subject to euthanasia and five horses with SAD. Of the horses that were euthanized, there were 106 horses that had a distal limb fracture, 59 horses with a proximal limb fracture and six horses with fracture associated with cervical or thoracic trauma. In addition, there were ten horses with a tendon or ligament injury and two horses with internal haemorrhage, which were euthanized. Five horses that had SAD were also classified as OFI; two horses with internal haemorrhage, two horses with cervical fracture and one horse with a fractured pelvis. Pulmonary lesions were also noted in the final diagnosis of 13 horses classified with OFI. These were EIPH in five horses, pulmonary oedema and congestion in four horses, EIPH plus pulmonary oedema and congestion in three horses and pulmonary oedema alone in one horse.

In addition to a final diagnosis including EIPH in 20/57 SCD horses and 13/188 OFI horses, a standardised system of grading of pulmonary lesions was employed in 34/57 (59.6%) post-mortems of horses with SCD and 129/188 (68.6%) of horses with OFI. This system graded pulmonary lesions attributed to pulmonary oedema, congestion, acute haemorrhage and chronic haemorrhage on a scale of 0–4, 0 being no lesions and 4 being severe lesions. Pulmonary lesions were observed in horses with both SCD and OFI and a summary of findings are presented in Appendix A.

### 3.3. Risk Factors Associated with SCD

The results of univariable logistic regression for each investigated variable are presented in Table 1. Of the eight variables investigated, the four retained in the multivariable model were sex, sire hemisphere of origin, exercise type and lifetime starts. Age and lifetime starts were correlated; Spearman *r* = 0.81, *p* < 0.001, supplementary Appendix A. Lifetime starts was retained as this variable had a greater impact on the model and age was discarded. Multivariable logistic regression results with age and lifetime starts retained are presented in Appendix A and with age retained and lifetime starts discarded are presented in Appendix A. A further three variables; hemisphere of origin, dam hemisphere of origin and season were discarded from the multivariable model. The results of multivariable logistic regression are presented in Table 2. In the multivariable model, exercise type at the time of fatality (racing versus training), sex (female, gelding or entire male) and lifetime starts were associated with SCD. Horses with SCD were more likely to die during training (OR 2.53, 95% CI 1.22–5.41; *p* = 0.01) and had fewer lifetime starts (OR 0.96, 95% CI 0.92–0.99; *p* = 0.02). Horses with SCD were less likely to be entire males compared to OFI (OR 0.37, 95% CI 0.14–0.89; *p* = 0.03).

### 3.4. Survival Analysis

The median (IQR) survival for horses with SCD was 3.6 (3.1–4.4) years and for horses with OFI was 4.5 (3.1–6.0) years. The Kaplan–Meier survival curves for horses with SCD and OFI are presented in Figure 2. Horses with SCD had a shorter survival than horses with OFI; hazard ratio (95% CI) 1.6 (1.1–2.3), *p* < 0.001.

### 3.5. Comparison of SCD Cases to Horses in the Australian Racing Population

There were 388,299 horses racing in Australia during 2007–2020. The frequency distribution of age at the time of fatality for horses with SCD and OFI compared to all racing horses is presented in Figure 3. The median (IQR) age at the time of fatality for horses with SCD was 3.0 (3.0–4.0) years, which was younger than the median age of horses racing in Australia; 4.0 (3.0–5.0) years, *p* = 0.007 and horses with OFI; 4.0 (3.0–6.0 years), *p* = 0.001. The median age of horses with OFI was not different to all racing horses.

## 4. Discussion

This study identified several key findings. Firstly, exercise type at the time of death, lifetime starts and sex were significantly associated with SCD fatality. Horses with SCD were more likely to die during training and had fewer lifetime starts than horses with other fatal injuries and there were fewer entire (uncastrated) males with SCD compared with other fatal injuries. Secondly, horses with SCD had shorter survival compared to horses with other fatal injuries. Finally, horses in the SCD group were younger than the Australian racing population and those with other fatal injuries, further supporting that these horses are vulnerable to fatalities early in their racing careers.

Sudden cardiac death is a distinct subgroup within sudden death. Sudden athletic death comprises all fatalities in which there is acute collapse and death in an otherwise healthy horse, during or immediately after exercise [3,24]. A diagnosis of SCD is restricted to horses with a diagnosis of likely fatal cardiac arrhythmia. Often, the diagnosis of likely fatal cardiac arrhythmia is reached in the absence of definitive evidence and following the exclusion of other known causes. The incidence of sudden death in horses is estimated to be 200 times the rate in human athletes [3]. Sudden athletic death accounts for 1–3 deaths per 10,000 starts in Thoroughbred racehorses and comprises approximately 13–25% of Thoroughbred flat racing fatalities [6,25,26,27]. Previous studies have investigated risk factors for sudden athletic death [10,14]. Ours is the first study to include horses in training when investigating risk factors for SCD. Sudden cardiac death occurred in association with training in three-quarters of the horses in our study. This finding highlights that many cases of SCD could be overlooked if focusing only on race day fatalities. Approximately half of all OFI in our study were sustained during training, which is similar to the previously reported frequency of 48–78% [16,19,27]. It is recognised that horses spend more time training but are proportionally more likely to sustain a musculoskeletal injury while racing [16,19,26,27,28]. Catastrophic musculoskeletal injury is strongly associated with volume of high speed exercise [29]. Most training exercise in Thoroughbred racehorses is performed at low-moderate intensity [30] and the cumulative volume of high speed exercise over one month of training approximates the volume of a single race [30]. Other than being of reduced intensity, training exercise is similar to racing exercise, although horses can be worked on different track surfaces and swimming and treadmill exercise are novel training methods. None of the horses in this study had a fatality associated with novel training methods. The effect of different types of training exercise on risk for SCD is yet to be investigated. The prevalence of arrhythmias increases with exercise intensity [31]. However, the results of our study showed that most SCD fatalities occurred during training, therefore exercise intensity appears not to be critically important in precipitating fatal arrhythmias in horses. This is in agreement with two studies in performance horses and racehorses, in which approximately half of all deaths occurred during training, with the remainder occurring in active competition [9,18]. A previous study identified that of sudden deaths occurring on race day, 57–65% of deaths occurred during a race with the remainder in the post-race period [5,14]. Clinically important arrhythmias occur commonly during both intense exercise and in the immediate post-exercise period [31,32,33,34]. In horses evaluated on a treadmill, higher-grade arrhythmias during peak exercise were associated with hypercapnoeia and hyperlactataemia [35]. In a separate study, complex cardiac arrhythmias were associated with poor performance and upper respiratory tract abnormalities [32]. The high prevalence of complex ectopic and re-entrant arrhythmias in the post-exercise period is thought to be caused by rapid changes in autonomic tone and metabolic derangements [24,32,33]. A single study has described electrocardiographic changes concurrent with SCD in a racehorse and found that complex ectopic and re-entrant arrhythmias progressing to ventricular fibrillation preceded death in that case [1]. In one large study of human athletes, it was recognised that only 61% of SCD occurred during exertion, with 13% occurring during sleep [13]. In the same study, sudden cardiac deaths occurring during exertion were more likely to have an underlying structural cardiac disease such as arrhythmogenic right ventricular cardiomyopathy or left ventricular fibrosis [13]. Sudden cardiac deaths occurring at rest were more likely to have negative post mortem findings and be attributed to sudden arrhythmic death syndrome [13]. Although not a specific goal of our research, very few deaths occurred at rest in horses in this sample, which is in agreement with other work in this species [26]. Further research is needed to understand the role of exertion in the development of fatal cardiac arrhythmias in horses.

The identification of horses being vulnerable to SCD early in their racing careers is a unique finding. Horses with SCD had fewer lifetime starts and shorter survival than horses with OFI and were younger than the racing population in Australia. Comparison of the frequency distribution of age to the racing population was recently used to investigate tibial and humeral stress fractures and highlighted that unique risk factors are appreciated for specific injuries within the umbrella of catastrophic musculoskeletal injury [36]. Whilst the risk factors identified in the present study are non-modifiable, they draw attention to the potential underlying causes of the syndrome of SCD in horses. Sudden cardiac death is not a diagnosis in itself but is the terminal event of multiple disorders. A small proportion (11%) of SCD horses in our study were definitively diagnosed with cardiomyopathy or myocarditis. However, other potential causes include congenital or inherited conditions, which would be expected to disproportionately affect younger horses. In humans, the underlying causes of SCD are broadly distinct in younger athletes from older athletes (≥35 years) [11]. Inherited or congenital cardiac diseases present almost exclusively in younger athletes and encompass the majority of SCDs in this group [11]. Such diseases include hypertrophic cardiomyopathy and arrhythmogenic right ventricular cardiomyopathy [11]. Molecular genetic screening can help to diagnose inherited channelopathies, long QT syndrome, catecholaminergic polymorphic ventricular tachycardia, and Brugada syndrome [11]. Coronary artery anomalies are another common cause of SCD in young athletes [37], and these are rarely reported in the horse [38]. Specific arrhythmic syndromes are rarely documented in horses, although ventricular pre-excitation syndromes have been reported and can be a cause for exclusion from exercise [24,39,40]. A genetic predisposition for sudden death was not found in Thoroughbred horses in the USA [9] and a recent study in Finnish trotters did not find an association between pedigree and sudden death [41]. In contrast to our results, increased age was associated with sudden death in one study of horses [14]. Whilst distinct to the findings in the present study, fatalities occurring in older horses could support the role of exercise-induced fibrosis and remodelling in the aetiology of SCD in horses. Myocardial fibrosis is commonly observed at post-mortem in trained racehorses [1,5,6,9,42]. In human athletes, myocardial fibrosis is more prevalent in older athletes than young athletes and is likely influenced by cumulative training volume [43,44]. Electrical remodelling is also appreciated in human athletes and can complicate the interpretation of electrocardiograms [45]. Remodelling of the AV node is shown to be influenced by training volume and AV block is particularly prevalent in trained racehorses [46]. Volumes of exercise undertaken by racehorses vary widely according to trainers [30] and could impact cardiac remodelling. Whilst our study found younger horses were at higher risk of SCD, the role and pathogenicity of electrical remodelling and fibrotic changes in the myocardium of horses remains to be characterised and could be important in some cases. Further study should also be directed at investigating potential congenital or inherited cardiac diseases in equine athletes.

Horseracing is relatively unique amongst competitive sports in that males and females undertake equivalent training and compete together in the same races. Registrations of male and female Thoroughbred racehorses in Australia are close to equivalent [23]. In human athletes, males are at 2–5 times higher risk of SCD than female athletes [13,47,48]. In human female athletes, lower participation rates at the elite level and lower prevalence of cardiac abnormalities capable of causing SCD are thought to account for the lower risk of SCD compared to males [47]. Our study found fewer entire male horses with SCD compared to horses with OFI. Previous studies of horses have identified that entire males are at higher risk of musculoskeletal injuries [15,26,27]. This is likely due to increased bodyweight in entire males resulting in greater musculoskeletal stress and higher risk for injury. The influence of gender on risk for equine fatalities could be a combination of effects on bodyweight and genetic and hormonal effects.

A limitation of this study was that horses were sampled from a single regional area, metropolitan Melbourne. This study design minimised bias, with all fatalities occurring in this region submitted for post-mortem examination. However, regional differences in environmental conditions such as weather and track surface, training practices and genetic influences might have impacted the results. Therefore, these findings might not be directly applicable to horses in other regions. A further limitation of our study was the difficulty in definitively classifying horses with SCD. Cardiac failure due to arrhythmia will often have negative post-mortem findings even when thorough gross and histopathological examination is conducted [1,5,41,49]. Up to 40% of sudden deaths in human athletes remain unexplained after comprehensive post mortem examination and disagreement between pathologists as to the cause of death is not uncommon [11,13,50,51]. Other studies investigating sudden death in horses have also reported on the difficulty of classifying horses with pulmonary lesions [6,10,14,41]. Consistent with other investigators [41], in our study horses with pulmonary lesions in which SCD could not reasonably be excluded were classified with SCD. This was based on the understanding that in many cases where pulmonary haemorrhage, congestion or oedema were attributed as the cause of death, such changes could have been incited by increased left atrial pressure as a result of cardiac failure [14,25]. Exercise induced pulmonary haemorrhage occurs in the majority of racing Thoroughbred horses [52] and was commonly identified in horses in this study classified with SCD. Exercise induced pulmonary haemorrhage, pulmonary congestion and oedema were also observed in some horses in this study that were subject to euthanasia due to an orthopaedic injury. This further supports the concept that pulmonary lesions might not be causative and can be observed in many horses dying suddenly or euthanized subsequent to exercise. Whilst fatal pulmonary haemorrhage has been implicated in sudden athletic death [5,6], the causal relationship between EIPH and sudden death is unclear [52]. Increased age and increased lifetime starts have been identified as risk-factors for EIPH [52,53,54]. The misclassification of some horses with fatal EIPH might have impacted on the results of our study. However, the disparity between recognised risk factors for EIPH and the risk factors for SCD identified by our study suggests that the potential misclassification of such horses is unlikely to have had a meaningful impact on our results. Another limitation of this study was that the control group, horses with OFI, was mostly comprised of horses with catastrophic musculoskeletal injury. Previous studies have consistently identified older horses, and entire males being at increased risk of musculoskeletal injury [15,29]. Therefore, the finding of SCD horses having fewer lifetime starts and being less likely to be entire males could be the reciprocal of the risk factors for musculoskeletal injury. A previous study investigating causes for SAD associated with racing concluded that risk factors for SAD were not uniquely different from all causes of fatality [14]. In our study, the identification of risk factors for SCD which were distinct from other causes of fatality was an important finding.

The identification of horses at risk of SCD prior to a clinical event remains challenging. Our study identified that the majority of these events occur during training and this raises the possibility of improved monitoring for cardiac disorders of horses as they commence and continue through training. Emerging technologies allow for enhanced monitoring of horses during exercise, with device capabilities including heart rate, ECG, velocity, stride length and stride frequency [55,56,57,58,59]. The sensitivity of these devices for identification of abnormalities of cardiac rhythm might be enhanced through the application of heart rate variability algorithms [60]. Further study is needed to determine the capability and application of such devices for screening of equine athletes.

## 5. Conclusions

We identified distinct risk factors for SCD in Thoroughbred racehorses incorporating data from both racing and training. Young horses, early in their racing careers were shown to be at increased risk of SCD and were more likely to die during training. This provides important new information and supports the potential role of inherited or congenital cardiac disorders in the pathogenesis of SCD in horses.

## Figures and Tables

**Figure 1 animals-12-01297-f001:**
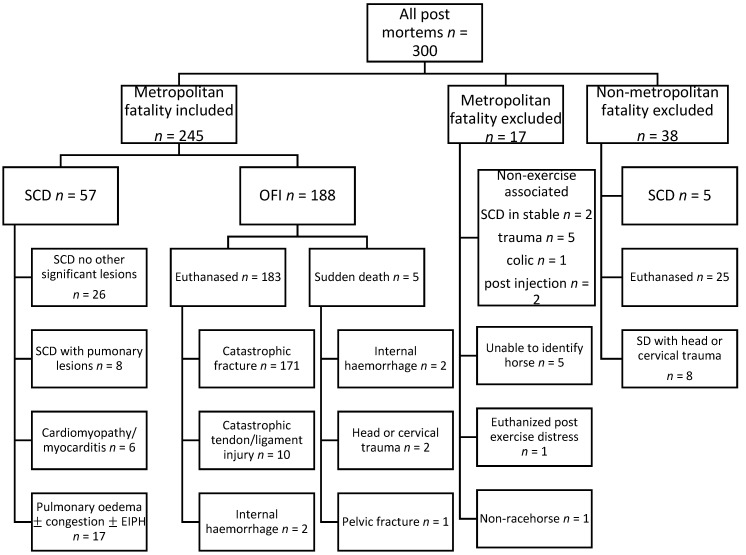
Flow chart summarising case selection from 300 post-mortems performed between 2007 and 2021. Non-exercise associated and non-metropolitan fatalities and non-racehorses were excluded, as were cases in which the identity of the horse could not be confirmed and a single case which was euthanized for post exercise distress. Sudden cardiac deaths comprised all sudden deaths in which no significant lesions were found and sudden deaths in which a diagnosis of cardiomyopathy/myocarditis was reached. Also included in the SCD group were cases with pulmonary lesions in which SCD could not reasonably be excluded. Other fatal injury (OFI) comprised all other fatalities during the study period.

**Figure 2 animals-12-01297-f002:**
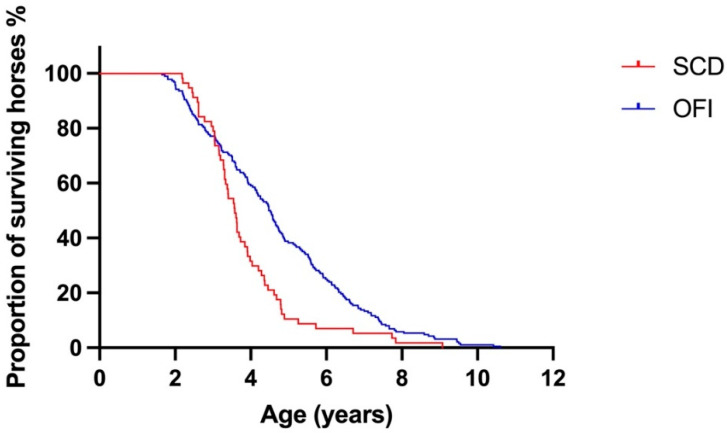
Kaplan–Meier survival curve for horses with sudden cardiac death (SCD) and other fatal injury (OFI), demonstrating non-proportional hazard. Overall, horses with SCD had a shorter survival than horses with OFI. Prior to 3.047 years, horses with OFI had shorter survival than horses with SCD. RMST, restricted mean survival time, was calculated for all time periods, before 3.047 years and after 3.047 years. At all time periods, life expectancy difference (LED), was 87.9 and life expectancy ratio (LER) was 1.23. Before 3.047 years, LED was −18.5 and LER was 0.92. After 3.047 years LED was 106.4 and LER was 1.81. Difference in median = 0.92 years (4.50 vs. 3.58). Hazard ratio = 1.6 (95% CI: 1.2–2.3), *p* < 0.001.

**Figure 3 animals-12-01297-f003:**
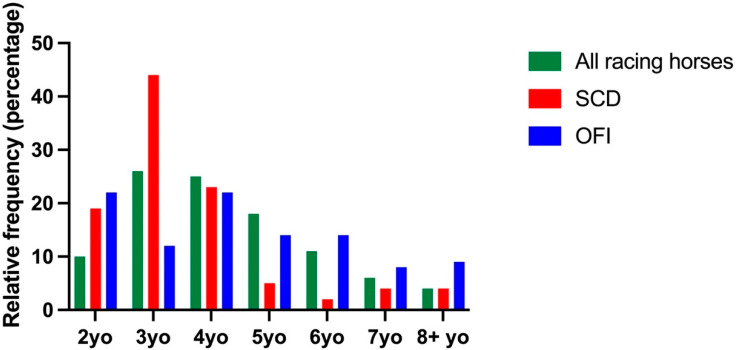
Frequency distribution of calendar age for all racing horses, horses with SCD and horses with OFI. Data for each group are presented as a percentage of the entire sample population.

**Table 1 animals-12-01297-t001:** Univariable logistic regression of relationship between each examined variable and SCD. Variables with *p* < 0.25 were submitted to the final multivariable model.

Variable Continuous, Median (IQR) Categorical, Proportion (%[95% CI])	SCD *n* = 57	OFI *n* = 188	β Coeff	Log Odds Ratio	OR 95% CI	*p* Value
Age (years)	3.58 (3.05–4.36)	4.50 (3.12–6.02)	−0.27	0.76	0.62–0.91	<0.001
Sex						
Gelding Female Entire male	27 (47[35–60]%) 21 (37[25–50]%) 9 (16[8–28]%)	94 (50[43–57]%) 48 (25[10–32]%) 46 (24[19–31]%)	−0.11 1.18 −2.90	0.90 3.24 0.05	0.50–1.63 1.76–6.03 0.00–0.26	0.73 <0.001 <0.001
Hemisphere of origin						
Southern hemi (REF) Northern hemi	53 (93[83–98]%) 4 (7[2–17]%)	171 (91[86–94]%) 17 (9[6–14]%)	−0.27	0.76	0.21–2.17	0.63
Sire hemisphere of origin						
Southern hemi (REF) Northern hemi	36 (63[50–75]%) 21 (37[25–50]%)	98 (52[45–69]%) 90 (48[41–55]%)	−0.45	0.64	0.34–1.16	0.14
Dam hemisphere of origin						
Southern hemi (REF) Northern hemi	48 (83[71–91]%) 9 (17[9–29]%)	155 (82[76–87]%) 33 (18[23–34]%)	−0.13	0.88	0.37–1.90	0.76
Exercise type						
Racing (REF) Training	15 (26[17–39]%) 42 (74[63–81]%)	106 (56[49–63]%) 82 (44[37–51]%)	1.29	3.62	1.91–7.16	<0.001
Lifetime starts	3 (0–9.0)	9 (0–22.75)	−0.05	0.95	0.91–0.97	<0.001
Season						
Summer Autumn Winter Spring	16 (28[18–41]%) 11 (19[11–31]%) 15 (26[17–39]%) 15 (26[17–39]%)	46 (24[19–31]%) 46 (24[19–31]%) 42 (22[17–29]%) 54 (29[23–36]%)	0.19 −0.30 0.22 −0.12	1.20 0.74 1.24 0.89	0.61–2.32 0.34–1.50 0.61–2.42 0.44–1.70	0.58 0.42 0.53 0.72

**Table 2 animals-12-01297-t002:** Multivariable logistic regression of relationship between each included variable and SCD.

Variable Continuous, Median (IQR) Categorical, Proportion (%[95% CI})	SCD *n* = 57	OFI *n* = 188	β Coeff	Log Odds Ratio	OR 95% CI	*p* Value
Sex						
Gelding (REF) Female Entire male	27 (47[35–60]%) 21 (37[25–50]%) 9 (16[8–28]%)	94 (50[43–57]%) 48 (25[10–32]%) 46 (25[19–31]%)	−0.04 −1.00	0.96 0.37	0.45–2.02 0.14–0.89	0.91 0.03
Sire hemisphere of origin						
Southern hemi (REF) Northern hemi	36 (63[50–75]%) 21 (37[25–50]%)	98 (52[45–69]%) 90 (48[41–55]%)	−0.28	0.76	0.39–1.44	0.40
Exercise type						
Racing (REF) Training	15 (26[17–39]%) 42 (74[63–81]%)	106 (56[49–63]%) 82 (44[37–51]%)	0.93	2.53	1.22–5.41	0.01
Lifetime starts	3 (0–9.0)	9 (0–22.75)	−0.04	0.96	0.92–0.99	0.02

Tjur’s r^2^ = 0.11, area under ROC curve (95% CI) = 0.73 (0.66–0.80), *p* < 0.0001. Hosmer-Lemeshow statistic 22.26, *p* = 0.005. Log-likelihood ratio (G squared) statistic 29.53, *p* < 0.0001.

## Data Availability

Due to confidentiality, data supporting this study is not publicly available.

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
