# Peer review of "Risk Factors for Exercise-Associated Sudden Cardiac Death in Thoroughbred Racehorses"

_animals, 2022, doi:10.3390/ani12101297_

Round 1

Reviewer 1 Report

The issue of racehorse death is important from both medical and economic points of view. Identifying the risk factors and possible predictors of racehore death and providing a statistic concerning the incidence of these events is of high value. Therefore I consider the paper to be of high interest and value.

However, there are a few issues that need to be addressed by the authors:

Firstly, the bibliographic indexes are not correctly included in the text and need revising.

Secondly, some termes need explanations such as "entire male".

  • lines 243-244 : "were less likely 243 to be entire males than horses with other fatal injuries" is a bit ambiguous.
  • Does training include particularly difficult activities that could promote and enhance the risk of SCD? Or could it represent a screening method for the risk of SCD in racehorses?
  • lines 362-364: "This further supports the concept that pulmonary le-362 sions might not be causative and can be observed in many horses dying suddenly or eu-363 thanized subsequent to exercise." - however, what is the impact of pulmonary lesions in racehorses diagnosed with SCD?

Author Response

Thank you very much for your valuable feedback on our manuscript. We believe we have addressed you concerns.

Bibliographic references have been checked and modified. To us they now appear correct.

Uncastrated has been added in brackets when the term entire male is first introduced.

The following sentence has been added at line 244: There were fewer entire (uncastrated) males with SCD than with other fatal injuries. 

The following has been added at line 269: Other than being of reduced intensity, training exercise is similar to racing exercise, although horses can be worked on different track surfaces and novel training methods include swimming exercise and treadmill exercise. None of the horses in this study had a fatality associated with novel training methods. The effect of different types of training exercise on risk for SCD is yet to be investigated. 

Screening for cardiac abnormalities during training would certainly be worthwhile and we have included this in the discussion at lines 511-523. "

The identification of horses at risk of SCD prior to a clinical event remains challenging. Our study identified that the majority of these events occur during training and this raises the possibility of improved monitoring for cardiac disorders of horses as they commence and continue through training. Emerging technologies allow for enhanced monitoring of horses during exercise, with device capabilities including heart rate, ECG, velocity, stride length and stride frequency [55-59]. The sensitivity of these devices for identification of abnormalities of cardiac rhythm might be enhanced through the application of heart rate variability algorithms [60]. Further study is needed to determine the capability and application of such devices for screening of equine athletes."

The impact of pulmonary lesions in horses diagnosed with SCD is not clear. It is possible that "such changes could have been incited by increased left atrial pressure as a result of cardiac failure", as stated in line 461. We have also added the statement that "Whilst fatal pulmonary haemorrhage has been implicated in sudden athletic death [5, 6], the causal relationship between EIPH and sudden death is unclear [52]."

Reviewer 2 Report

The authors have provided a well written and interesting study on risk factors for sudden cardiac death in thoroughbred racehorses. A retrospective case-control study with 57 cases and 188 controls was conducted and risk factors were studied using univariate and multivariate logistic regression analyses. The authors conclude that young horses have an increased risk of sudden cardiac death and sudden cardiac death occurs more during training. 

It is very interesting that many horses die from a sudden cardiac event during training and not during races. Authors report all deaths within one hour after training/race were considered exercise related. To be the devil's advocate, horses train more often than they race so chances of having an event during training is higher than having an event during less frequent races. What are the authors' thoughts about this? 

Do the authors have any data on recovery time between races and training? Can it be that horses that are allowed to recover longer have less risks? Also the authors report that the volumes of exercises vary widely. Do you have any data to study what volume of exercise has the lowest risk of SCD? 

The authors report a Kaplan-Meier survival curve and a hazard ratio. However, since the Kaplan-Meier curves cross, it is not allowed to calculate a hazard ratio. See for example https://doi.org/10.1136/bmj.j2250

Please define EIPH when first used. 

Typo in introduction line 68 'incresed' 

Author Response

We agree that horses spend proportionally more time training than racing, and therefore have more opportunity to experience an adverse event during training. However, horses usually undertake sub-maximal exercise intensity during training and maximal exercise intensity during racing. We have stated in lines 341-343 that "It is recognised that horses spend more time training but are proportionally more likely to sustain a musculoskeletal injury while racing [16, 19, 26-28]. " We have also added at lines 346-347 "Other than being of reduced intensity, training exercise is similar to racing exercise, although horses can be worked on different track surfaces and novel training methods include swimming exercise and treadmill exercise. None of the horses in this study had a fatality associated with novel training methods. The effect of different types of training exercise on risk for SCD is yet to be investigated." We include the following at line 349 "The prevalence of arrhythmias increases with exercise intensity [31]." Exercise intensity has previously been shown to be important in precipitating musculoskeletal injury and higher grade arrhythmias are also observed more frequently in horses exercising at higher intensity. The findings of our study suggest that although higher grade arrhythmias are observed more frequently at higher intensities, SCD, presumably due to fatal cardiac arrhythmia, can also occur during submaximal training exercise. Therefore we concluded that "Exercise intensity appears not to be critically important in precipitating sudden cardiac death in horses. " lines 21-22.

The effect of recovery time between races and training would be interesting to investigate. However we only had access to the racing records and the training record on the day of the fatality for horses in our study. It is possible that longer recovery could reduce risk and this might be true for musculoskeletal injury also. As yet we do not have any data on what volume of exercise is lowest risk for SCD.

Thank you for highlighting the non-proportional hazard in the Kaplan-Meier survival curve. We have included the extra description with restricted mean survival time prior to 3.047 years and after 3.047 years and the LED and LER at these time points as suggested by your reference. Although the curves cross early in the data set, we believe that our results are still meaningful, as the curves cross early in the series when there are few events. As event rate increases, the curves become more distinct.

EIPH has been defined.

Typo has been corrected.

Reviewer 3 Report

Please, see the pdf attached.

Author Response

Thank you very much for your review which has helped to improve the manuscript. We believe we have addressed your concerns.

The position of the reference has been modified throughout the manuscript.

1. Introduction

The following sentences have been modified to improve clarity. "Inherited or congenital cardiac diseases, including inherited cardiomyopathies and ion channelopathies, present almost exclusively in younger athletes and encompass the majority of SCDs in this group [11]. Therefore these are the primary cause of SCD in young athletes in the absence of acquired cardiovascular disease or training induced cardiovascular abnormalities. "

The following sentences have been moved down lines 68-71 and now follow the statement on cumulative volume of exercise to improve clarity in describing the pathophysiology "In human athletes, structural and electrical cardiac remodelling accompanies increased volumes of exercise training but their significance in the pathogenesis of arrhythmias and SCD is not clear [12, 13]. The underlying pathology associated with SCD is not well established in horses, although myocardial inflammation and fibrosis, chamber dilation and trabecular and myocardial hypertrophy have been described [1, 5, 9]."

This sentence has been reworked to make it more clear that previous studies have found fatalities occur more commonly in training. Lines 72-73 "Approximately 53-60% of sudden death fatalities in racehorses and horses engaged in other sport disciplines occur in training, supporting the concept that fatalities regularly occur at sub-maximal exercise intensity [9, 18]."

The following statement at Line 63 describes the shared risk factors for sudden athletic death and catastrophic musculoskeletal injury. "Increased age and race distance, along with racing in summer, have been identified as risk factors for race-associated SAD in Thoroughbred horses [14]. These risk factors are shared with catastrophic musculoskeletal injury [15-17]."

2. Materials and methods

Table 1 has been deleted and the sentence moved as suggested.

3. Results

The scale for pulmonary lesions has been added.

This sub heading has been changed as suggested.

We found that age was not associated with SCD in the multivariable model. It is our view that the multivariable model included in the main body of the manuscript is the most appropriate model for this dataset. Age and lifetime starts showed collinearity and therefore we think it was not appropriate to include these variables together in the model in the main manuscipt. We included both of the alternative models in the supplementary data, table 2 showing the effect of age and lifetime starts together and table 3 showing the effect of age without lifetime starts. It was our intention to include these tables in the interests of transparency. However if the editors and reviewers feel strongly that these should be removed we are happy to do so.

4. Discussion

"and those with other fatal injuries" has been added at line 348

The following has been added to explain the effect of sex at lines 449-453 "In human female athletes, lower participation rates at the elite level and lower prevalence of cardiac abnormalities capable of causing SCD are thought to account for the lower risk of SCD compared to males [47]. Our study found fewer entire male horses with SCD compared to horses with OFI. Previous studies of horses have identified that entire males are at higher risk of musculoskeletal injuries [15, 26, 27]. This is likely due to increased bodyweight in entire males resulting in greater musculoskeletal stress and higher risk for injury. We also state in the discussion at lines 495-497 "Previous studies have consistently identified older horses, and entire males being at increased risk of musculoskeletal injury [15, 29]. Therefore, the finding of SCD horses having fewer lifetime starts and being less likely to be entire males could be the reciprocal of the risk factors for musculoskeletal injury."

Tables include the proportion (%) and 95%CI for categorical variables and the median and IQR for continuous variables. This information is provided in the top left corner of the table.